# FROM INFORMATION BOTTLENECK TO ACTIVATION NORM PENALTY

## ABSTRACT

Many regularization methods have been proposed to prevent overfitting in neural networks. Recently, a regularization method has been proposed to optimize the variational lower bound of the Information Bottleneck Lagrangian. However, this method cannot be generalized to regular neural network architectures. We present the activation norm penalty that is derived from the information bottleneck principle and is theoretically grounded in a variation dropout framework. Unlike in previous literature, it can be applied to any general neural network. We demonstrate that this penalty can give consistent improvements to different state of the art architectures both in language modeling and image classification. We present analyses on the properties of this penalty and compare it to other methods that also reduce mutual information.

## 1 INTRODUCTION

Neural networks have been applied to many domains and are able to solve complex tasks such as image classification or machine translation, achieving higher performance than other families of learning algorithms. However, most neural networks used in these tasks are over-parameterized, and they are operating on inputs in high-dimensional space, thus prone to overfitting.

For the task of supervised learning, the goal is to find a mapping from noisy input $X$ to a set of corresponding labels $Y$. In the perspective of information theory, neural networks construct a mapping $S$ and then decode from the resulting representation $S(X)$ and obtain hypothesis $\hat{Y}$, forming the following Markov Chain: $Y \rightarrow X \rightarrow S$. By data processing inequality, $I(Y; X) \geq I(Y; S(X))$. So if $S$ captures the sufficient statistics of $X$, we have $I(Y; X) = I(Y; S(X))$ (Cover & Thomas, 2012). However, a trivial solution would satisfy this constraint: an identity mapping of $X$. To avoid this, we further require the minimum sufficient statistics of $X$ to satisfy $T^* = \arg\min_{I(Y;X)=I(Y;S(X))} I(S(X); X)$. The minimum sufficient statistics should only capture the most relevant features in $X$. Intuitively, being able to compute $T$ exactly would solve the overfitting problem.

However, the minimum sufficient statistics does not exist for general distributions. Tishby et al. (2000) relaxed the requirement and turn the problem into the Information Bottleneck Lagrangian: minimize $\beta I(X; T) - I(Y; T)$. We can work out a solution for $P(T|X)$ if $P(X, Y)$ is known. We would also obtain a solution if we have deterministic mapping between $T$ and $X$, namely for $I(X; T) = H(T) - H(T|X)$, $H(T|X) = 0$, then minimizing the mutual information becomes minimizing the entropy of $T$. Since we don't know the distribution of $T$, we are not able to penalize towards this objective either.

So we can only apply information bottleneck penalty in a principled way for probabilistic framework where $T|X$ is a specified or known distribution. Alemi et al. (2016) has proposed a variational lower bound of the information bottleneck objective when $T|X \sim \mathcal{N}(\mu(X), \text{diag}(\sigma^2(X)))$ where both $\mu$ and $\sigma$ are estimated by a neural network.

We first extend Alemi et al. (2016)'s work naively to recurrent neural network. Then instead of relying on mean field approximation variational inference, which is not widely applicable to general neural network architecture, we extend the variational approximation of the information bottleneck objective to any neural network with dropout, which is shown to be mathematically equivalent to the lower bound of a Gaussian Process (Gal & Ghahramani, 2015). We present an information

bottleneck penalty that has a very simple equivalence in a neural network with dropout. From additional demonstration by Gal & Ghahramani (2016), we can easily extend our case in recurrent neural networks as well.

We validate this penalty on language modeling and image classification, we observe improvements over all near state of the art baselines. Finally, we show preliminary comparisons between this penalty and its variations.

## 2 PRELIMINARIES

### 2.1 VARIATIONAL DROPOUT

Neural networks trained with dropout can be viewed as approximating a Gaussian Process (Gal & Ghahramani, 2015). We sample $\boldsymbol{w}_k \sim P(\boldsymbol{w})$ and $b_k \sim P(\boldsymbol{b})$ for $K$ hidden units in a single layer. Define $\phi(\boldsymbol{x}, \boldsymbol{W}, \boldsymbol{b}) = \sqrt{\frac{1}{K}}\sigma(\boldsymbol{W}x + \boldsymbol{b})$, this is the activation output of a single network layer, and $\sigma$ is a non-linear function. The training dataset contains $N$ samples, we can define $\Phi = [\phi(\boldsymbol{x}, \boldsymbol{W}, \boldsymbol{b})]_{n=1}^N$, with $\Phi \in \mathbb{R}^{N \times K}$. We can then form a valid covariance matrix by computing the outer product of our layer output $\hat{K}(X, X) = \Phi\Phi^T$.

We define the following Gaussian Process:

$$
\begin{aligned}
\mathbf{T}|\mathbf{X}, \mathbf{W}, \boldsymbol{b} &\sim \mathcal{N}(0, \hat{K}(\mathbf{X}, \mathbf{X})) \\
\mathbf{Y}|\mathbf{T} &\sim \mathcal{N}(\mathbf{T}, 0 \cdot \mathbf{I}_N) \\
c_n|\mathbf{Y} &\sim \text{Categorical}\Big(\frac{e^{y_{nd}}}{\sum_{d'} e^{y_{nd'}}}\Big)
\end{aligned}
\tag{1}
$$

This setting can be extended to recurrent neural network. Gal & Ghahramani (2016) pointed out as long as the dropout mask stays the same across time steps, it will give the same weight and bias realisation. It is shown that optimizing such neural network with dropout is equivalent to optimizing the evidence lower bound of the Gaussian Process above.

The dropout mask is applied to weight matrices. Define $\mathbf{W} : Q \times K$, and $\boldsymbol{x} : 1 \times K$, a row vector. The prior of weight vector can be defined as a Gaussian mixture with probability p. $\boldsymbol{W} = [\boldsymbol{w}_i]_{i=1}^Q$. The sampling of weight matrix $\mathbf{W}$ can be reparameterizedm by using a Bernoulli random vector $\boldsymbol{z} : Q$, with $\boldsymbol{z}_i \sim \text{Bernoulli}(p)$ in (2) and let $\mathbf{M} = [m_i]_{i=1}^Q$.

$$
\begin{aligned}
q(\boldsymbol{w}_i) &= p\mathcal{N}(\boldsymbol{m}_i, \boldsymbol{\sigma}^2 \mathrm{I}) + (1 - p)\mathcal{N}(0, \boldsymbol{\sigma^2}\mathrm{I}) \\
\mathbf{W} &= \boldsymbol{z}(\mathbf{M} + \boldsymbol{\sigma\epsilon}) + (1 - \boldsymbol{z})\boldsymbol{\sigma\epsilon} \\
\boldsymbol{b} &= \boldsymbol{m} + \boldsymbol{\sigma\epsilon}
\end{aligned}
\tag{2}
$$

We can write the optimization objective as follows, note that this objective requires the weight and bias dropout mask sampled per example. For a recurrent neural network in language modeling, we can compute the average this loss computed across the time step.

$$
\mathcal{L}_{\text{GP}-\text{VI}}(\theta) = -\sum_{n=1}^N \log p(\mathbf{Y}_n|\mathbf{X}_n, \mathbf{W}_n, \mathbf{b}_n) + D_{KL}(q_\theta(\mathbf{W}, \mathbf{b})||p(\mathbf{W}, \mathbf{b}))
\tag{3}
$$

### 2.2 VARIATIONAL INFORMATION BOTTLENECK METHOD

Tishby Tishby et al. (2000) proposed the training objective of information bottleneck method assuming the algorithm forms a markov chain: $Y - X - T$, where $T$ is the learned minimum sufficient statistics of $X$.

$$
\max_{p(t|x), p(y|t), p(t)} I(Y; T) - \beta I(X; T)
\tag{4}
$$

Alemi et al. (2016) derived a variational lower bound by substituting $p(Y|T)$ with $q(Y|T)$, and by $D_{KL}(p(Y|T)||q(Y|T)) \geq 0$ and non-negativity of discrete entropy, we derive a variational lower bound for $I(Y;T)$.

$$I(Y;T) \geq \int p(y,t)q(y|t) - H(Y) \geq \int dydt p(y,t)q(y|t)dydt \qquad (5)$$

Similarly, since we do not have the joint probability distribution of data $P(X,Y)$, we can't compute $p(t) = \int dx p(t|x)p(x)$ . Instead, we choose the distribution of the representation to be an arbitrary distribution, thus $D_{KL}(p(T)||r(T)) \geq 0$:

$$I(X;T) \leq \int p(x)p(t|x) \log \frac{p(t|x)}{r(t)} dxdt \qquad (6)$$

So the complete variational information bottleneck method objective can be approximated by Monte Carlo approximation without knowing the true joint distribution of $X$ and $Y$.

$$I(Y;T) - \beta I(X;T) \approx \frac{1}{N} \sum_{i=1}^{N} \mathbb{E}_{\epsilon \sim p(\epsilon)}[-\log q(y_n|f(x_n,\epsilon))] + \beta D_{KL}(p(T|x_n)||r(T)) \qquad (7)$$

The first term that maximizes $I(Y;T)$ is the normal negative log-likelihood loss that is used to train classification algorithms, and the second term that minimizes $I(X;T)$, in the implementation, is often simplified as the KL divergence between $p(T|X)$ and any distribution $r(T)$. We delegate more detailed explanation to Alemi et al. (2016).

## 3  METHOD

### 3.1  BAYESIAN INFORMATION PENALTY

We naively extend Alemi et al. (2016)'s result to a recurrent neural network by adding a stochastic layer on top of the deterministic computation. Our implementation is a simplified version of Stochastic Recurrent Neural Network (SRNN) by Fraccaro et al. (2016). The goal is to compute a conditional density function $p_\theta(z_t|z_{t-1}, h_t)$ for a hidden state $z_t$. We can specify that $p_\theta(z_t|z_{t-1}, h_t) = \mathcal{N}(\mu_t, \text{diag}(\sigma))$.

In Fraccaro et al., a neural network computes $\log \sigma_t$ for each time step given $z_{t-1}$ and $h_t$. In our setup, $\sigma$ is a learnable parameter that is adjusted by the overall optimization objective. Instead of viewing it as the uncertainty on the data, treating $\sigma$ as a parameter allows the network to signal its own confidence on the prediction it makes, irrelevant to a particular datapoint. It can also be treated as adding auto-adjusting noise to the output of the layer, a form of automatic regularization. Define $G$ to be the LSTM gating function parameterized by $\omega = \{W_x, U_h\}$.

$$
\begin{aligned}
[i,f,g,h] = G_\omega(h_{t-1}, x_t) \qquad & \mu_t = o \odot \tanh(c_t) \\
c_t = f \odot c_{t-1} + i \odot g \qquad & h_t \sim \mathcal{N}(\mu_t, \text{diag}(\sigma^2))
\end{aligned}
\qquad (8)
$$

$$\mathcal{L}_i^t(\theta) = -\mathbb{E}_{q_\theta(z_t|x_{<t})}[\log p(x_t|z_t)] + D_{KL}(q_\theta(z_t|x_{<t})||p(z_t|x_{<t})) \qquad (9)$$

By forming a variational evidence lower bound, the typical Bayesian objective can be written as (9) for each sequence $i$ of at position $t$. In practical optimization, we assume the true posterior comes from a spherical Gaussian distribution $p(z_t|x_{<t}) \sim \mathcal{N}(0, I)$. It is easy to observe that the training objective of a Bayesian recurrent neural network already contains a information bottleneck objective. The first term can be thought of as the varitional bound from $I(Y;T)$ and the second term corresponds to the information bottleneck penalty from $I(X;T)$.

However, this approach suffers from the same problem as in Alemi et al. (2016)'s derivation: it limits the covariance matrix of the multivariate Gaussian to be diagonal, because the variational lower

bound is derived from the mean-field approximation. We know that neural network representation obtains its expressiveness from the distributed nature of each layer, thus forcing the output of neural network $T \sim \mathcal{N}(\mu, \text{diag}(\sigma^2))$ is desirable. We would like to apply information bottleneck objective to any general neural network.

## 3.2 ACTIVATION NORM PENALTY

Gal & Ghahramani (2015)'s demonstration allows us to apply information bottleneck objective to a Gaussian Process that has the kernel function parameterized and learned by a neural network. Optimizing the variational lower bound of this Gaussian Process with added information bottleneck penalty $I(X;T)$ is equivalent to optimizing the same loss with a regularization term on the underlying neural network with dropout. We refer to this regularization penalty as the activation norm penalty.

The first part of information bottleneck objective $I(Y;T)$ is equivalent to running the neural network with dropout and optimizing its negative log likelihood. The second part of the objective is to minimize on $D_{KL}(p(T|X,W,b)||r(T))$, where $r(T) \sim \mathcal{N}(0, \boldsymbol{I})$. According to the Gaussian Process we defined, we now have a distribution for $T$, and we can evaluate this KL divergence on these two terms, and $\Sigma_0 = \hat{K}(\boldsymbol{X}, \boldsymbol{X})$. We have an exact formula for the KL divergence between two k-dimensional multivariate Gaussian distributions given their sufficient statistics, so we can compute $I(X;T)$.

$$D_{\text{KL}}(p(\mathbf{T}|\mathbf{X}, \mathbf{W}, \mathbf{b})||r(\mathbf{T})) = \frac{1}{2}\{\text{tr}(\Sigma_1^{-1}\Sigma_0) + (\mu_1 - \mu_0)^{\text{T}}\Sigma_1^{-1}(\mu_1 - \mu_0) - k - \ln\frac{|\Sigma_1|}{|\Sigma_0|}\}$$

$$= \frac{1}{2}\{\text{tr}(\hat{K}(\mathbf{X}, \mathbf{X})) - \ln|\hat{K}(\boldsymbol{X}, \boldsymbol{X})|\} \quad (10)$$

$$\text{tr}(\hat{K}(\mathbf{X}, \mathbf{X})) = \frac{N}{K}\sum_i^N[\sigma(\mathbf{W}^T\boldsymbol{x}_i + \boldsymbol{b})]^2$$

Our main result can be seen in the first term $\text{tr}(\hat{K}(\mathbf{X}, \mathbf{X}))$, which is equivalent to a $L_2$ norm penalty over the output of the final non-linear layer. We refer to it as the activation norm penalty (ANP). We define $\phi$ to be the entire neural network, and regularizing on the activations of the final layer is equivalent to minimizing $I(X;T)$ on the corresponding Gaussian Process. We additionally use a hyperparameter $\eta$ to control the strength of this regularizer.

$$\mathcal{L}_{\text{GP}-\text{ANP}}(\theta) = -\mathcal{L}_{\text{GP}-\text{VI}}(\theta) + D_{\text{KL}}(P(\mathbf{T}|\mathbf{X}, \mathbf{W}, \mathbf{b})||r(\mathbf{T}))$$

$$\approx -\mathcal{L}_{\text{GP}-\text{VI}}(\theta) + \eta\frac{N}{K}\sum_i^N[\phi(x)]^2 \quad (11)$$

It is important to note that this penalty is applied before $\boldsymbol{Y}|\boldsymbol{T} \sim \mathcal{N}(\boldsymbol{T}, 0 \cdot \boldsymbol{I}_N)$, which corresponds to the linear transformation $\hat{\boldsymbol{y}} = W^T\phi(x)$ prior to the softmax normalization.

As for the determinant of the covariance matrix of Gaussian Process, we cannot easily evaluate or derive its gradient, so we do not include it in our computation. We are able to show the possibility for the existence of its determinant: $\hat{K}$ is formed by outer product of two matrices $\Phi$ of $N \times K$, so $\text{rank}(\hat{K}) \leq \text{rank}(K)$, and since $K$ is the number of hidden units and $N$ is the batch size in most neural networks, $K >> N$, so it is possible for $\hat{K}$ to be a non-degenerate semi-definite matrix.

## 4 RELATED WORK

There exists other information-theoretic regularization methods for neural network. Pereyra et al. (2017) incentivize the network to increase the entropy of the label distribution $p_\theta(\boldsymbol{y}|\boldsymbol{x})$. Achille & Soatto (2017) derived a information bottleneck duality between weights and output then proved their bound is tight for the one-layer situation, thus showing that minimizing through weight decay penalty would decrease $I(X;T)$.

Similarly, in an effort to increase neural network's ability to generalize, Kawaguchi et al. (2017) derived a regularizer that is the $L_\infty$ norm of the output of the neural network, and observed improvements on MNIST and CIFAR-10. This penalty is very similar to Pereyra et al. (2017)'s attempt to increase entropy of the output distribution.

Our method differs from these previous methods in significant ways. We follow the information bottleneck principle which views the network as two parts: an encoder and a decoder. An encoder would be the entire network up to the last layer, and a decoder is the linear projection from the hidden space to the label space, where we use softmax operation to normalize these outputs into a probability distribution. Previous methods have only applied various penalty to the output of the decoder, but information bottleneck shows that the regularization penalty should be applied to the output of the encoder.

## 5 EXPERIMENTS

### 5.1 LANGUAGE MODELING

In order to demonstrate the general applicability of our regularization method, we apply the penalty to a recurrent neural network with LSTM units as well as SRU units (Lei & Zhang, 2017).

**Penn Treebank**   We train networks for word-level language modeling on the Penn Treebank dataset using the standard preprocessed splits with a 10K size vocabulary (Mikolov, 2012). The training set is composed of 929,590 tokens; the validation set contains 73,761 tokens, and the training set 82,431 tokens.

To train our language model, we set the initial learning rate at 1.0 and batch size to be 20. We anneal the learning rate when the perplexity in validation set is higher than previous best validation perplexity and restore to the previous best model . We tune the learning rate decay and maximum number of decays according to the validation set. We find setting learning rate decay factor to 0.7 and maximum number of decay to 15 to be the best hyperparameter for various settings. We set the batch size at 20, and initialize the parameters uniformly.

In our baseline 2-layer LSTM model, we only include feedforward dropout, and set the hidden states size to 512 and 1500, similar to Zaremba et al. (2014). In our medium-sized experiment setting, we compare regularization or data augmentation techniques such as shared embedding (Inan et al., 2016; Press & Wolf, 2016), data noising (Xie et al., 2017), and our method. We also build a base LSTM model with variational dropout. We sample two separate dropout masks for the feedforward direction and recurrent direction, and in addition to embedding dropout, we generate a skip-input dropout mask described by Gal & Ghahramani (2016), which skips timesteps. We set this skip-input dropout rate to be 0.2, input embedding dropout to be 0.5, and feedforward dropout rate to be 0.5

| Medium-sized models | | | |
|---|---|---|---|
| Model | Validation | Test | Δ |
| LSTM | 84.3 | 80.4 | — |
| LSTM + Data Noising | 79.9 | 76.9 | 3.5 ↓ |
| LSTM + Shared Embedding | 80.0 | 76.7 | 3.7 ↓ |
| LSTM + Activation Norm Penalty | 79.6 | **76.3** | **4.1** ↓ |
| VD-LSTM | 78.5 | 74.9 | — |
| VD-LSTM + Activation Norm Penalty | 77.6 | **73.9** | 1.0 ↓ |
| Bayesian Information Penalty | 83.8 | 80.7 | — |
| Bayes By Backprop (Fortunato et al., 2017) | 78.8 | 75.5 | — |

Table 1: Medium-model perplexity on Penn Treebank with different types of medium sized models. We report the improvement ↓ (a decrease in perplexity) and compare activation norm penalty with other regularization methods

| Large-sized models | | | |
|---|---|---|---|
| Model | Validation | Test | Δ |
| LSTM | 81.6 | 77.5 | — |
| LSTM + Data Noising | 76.2 | 73.4 | 4.1 ↓ |
| LSTM + Activation Norm Penalty | **76.7** | **72.8** | **4.7** ↓ |
| VD-LSTM | 74.4 | 71.2 | — |
| VD-LSTM + Activation Norm Penalty | **73.0** | **69.6** | **1.6** ↓ |
| SRU 5 Layer | 66.2 | 62.3 | — |
| SRU 5 Layer + Activation Norm Penalty | **64.5** | **61.3** | **1.0** ↓ |
| Variational RHN (Zilly et al., 2016) | 71.2 | 68.5 | — |
| NAS with base 8 (Zoph & Le, 2016) | — | 67.9 | — |

Table 2: Full-model perplexity on Penn Treebank. Activation norm penalty outperforms other regularization methods and can improve the perplexity on the state of the art architecture.

to our VD-LSTM model. We set recurrent dropout rate to be 0.2 for the medium and 0.5 for the large-sized model. We find the best $\eta$ to be 0.0001 for the medium, and 0.0002 for the large model.

For our SRU model, we fixed the random seed and run with the original training settings[1], except with batch size 64. Even though the baseline we obtained has a higher perplexity than reported in Lei & Zhang (2017), the optimization procedure and hyperparameters are set as default and fixed for both models, so it is a fair comparison. The best $\eta$ we find is $10^{-8}$.

**WikiText-2**   We also examine our model on Wikitext-2 (Merity et al., 2016). It is pre-split into train, validation, and test sections, each contains 2M, 217k, 245k, which is about twice as large as the PTB dataset. It is introduced as a more challenging dataset than simpler, more explored PTB for building language models.

Since our method is to mitigate overfitting, we want to apply the norm penalty to a model that exhibits overfitting behaviors. It is important to note PTB is a better corpus for regularization methods because models tend to exhibit large train/validation perplexity gap. We trained a full model using vanilla LSTM with feedforward dropout, and with the same set of hyperparameters we used for training on PTB.

| Model | Validation | Test |
|---|---|---|
| Variational LSTM | 101.7 | 96.3 |
| Pointer Sentinel LSTM | 84.8 | 80.8 |
| LSTM + Activation Norm Penalty | 94.4 | 89.7 |
| LSTM + Data Noising | 92.9 | 88.1 |
| LSTM + ANP + Data Noising | **89.1** | **84.5** |

Table 3: Single-model perplexity on Wikitext-2 with different types of models and regularization methods. We report the variational LSTM perplexity from Merity et al. (2016). When combining a regularization technique with data noising, we are still able to achieve a comparable performance to models that are more expressive.

## 5.2   IMAGE CLASSIFICATION

We examine our regularization in CIFAR-10 and CIFAR-100. We directly used publicly available implementation of Wide-ResNet [2] (Zagoruyko & Komodakis, 2016), since our regularization can be easily applied to any model.

---

[1] https://github.com/taolei87/sru

[2] https://github.com/meliketoy/wide-resnet.pytorch

Different from the report from Zagoruyko & Komodakis (2016), but similar to the findings of He et al. (2016), we observed a detrimental effect of dropout to image classification. So we applied activation norm penalty directly on the last layer of the model without dropout.

For the speed of training, we construct a model of 28 layers with the widen-factor of 10. We conducted a search for the activation norm penalty hyperparameter $\eta$ of and found the best $\eta$ for both datasets is $10^{-6}$. We also fixed the random seed.

**CIFAR-10**    CIFAR-10 contains image of 32x32 with 3 RGB channels. The training set contains 50k images, and the test set has 10k images. We used the best hyperparameter and data preprocessing setting provided by the model. We train all the models with default weight decay setting. Result is percentage in accuracy on test set.

**CIFAR-100**    CIFAR-100 provides images that are similar to CIFAR-10 but contains 500 images for the training of each class, and 100 images for each class in the test. We apply the penalty after the final max pooling layer.

| CIFAR-10 | | |
|---|---|---|
| Model | Test | $\Delta$ |
| Wide-ResNet | 96.19 | — |
| Wide-ResNet + Activation Norm Penalty | **96.41** | +0.22 |
| CIFAR-100 | | |
| Wide-ResNet | 80.78 | — |
| Wide-ResNet + Activation Norm Penalty | **81.56** | +0.78 |

Table 4: Wide-Residual network accuracy on CIFAR-10 and CIFAR-100.

## 6    ANALYSIS

**The Effects of Activation Norm Penalty**    Neural network naturally reduces mutual information or any type of linear or non-linear correlation between input and the final output through a series of non-linear activation function, pooling, and dropout. All of these contribute to our final objective of minimizing $I(X;T)$.

We find that in Table 1, the penalty is most effective to the recurrent neural network model without recurrent dropout. We are able to see that for LSTM with simple feedforward dropout, activation norm penalty even outperforms other widely-used methods like shared embedding.

However, the effectiveness of this method diminishes when dropout is applied aggressively in recurrent neural network, since intuitively dropout will decrease correlation between input and output. We also observe from Table 4 the improvements on bottleneck architectures (used in Wide-ResNet) are also small due to the fact these architectures already reduce mutual information implicitly without the need of external penalty.

We also observe improvements in image classification without dropout. Even though our theory fails to explain this phenomenon, it would indicate that there exists other explanations for the additional benefits of the activation norm penalty. One such explanation could be that this regularized loss function is effective in stabilizing neural networks as it forces them to produce hidden states from a limited region, improving the performance of final softmax linear projection. The importance of norm stabilization on hidden states has been argued by Krueger & Memisevic (2015). This way a neural network can train its weights to operate on a less variable region of hidden states.

**Effects on Layers of LSTM**    Even though $L_2$ norm gives us a good KL divergence interpretation and is connected with information bottleneck. Other hypotheses or theoretical frameworks might exist that can justify the using of other types of norm penalty. So we experimented on different types of norms applied to the last layer of medium size models on Penn Treebank.

Since all our models are using LSTM unit, we also experiment on the effect of placing $L_2$ norm penalty over cell states or hidden states and on different layers. So far, applying $L_2$ norm penalty over the last layer still yields the best performance.

| Model | Validation |
|---|---|
| $L_1$ norm | 84.4 |
| $L_\infty$ norm | 84.1 |
| $L_2$ norm | **79.6** |

(a) Validation perplexity on types of norm

| Model | Validation |
|---|---|
| Layer 1 - $c_t$ | 84.9 |
| Layer 1 - $h_t$ | 84.6 |
| Layer 2 - $c_t$ | 81.2 |
| Layer 2 - $h_t$ | **79.6** |

(b) Validation perplexity on parts of LSTM

Table 5: Validation perplexity on exploring the activation norm penalty technique

The result in Table 5 fits with our understanding, an LSTM unit can be considered as having 2 layers of computation, first $c_t$ is computed, then $h_t$ is computed. Once we decide to compress the source representation, in a neural network Markov chain $X - h_1 - h_2 - ... - h_n$, by data processing inequality $I(X; h_1) \geq I(X; h_{i>2})$, we cannot recover lost information in a later stage of processing.

**Difference with Weight Decay**   Even though the norm of activation is naturally tied with the norm of the weight matrices, since our penalty is applied after activation or other nonlinear transformation of the input such as pooling, it is not equivalent to weight decay. We are able to observe additive effect when used with weight decay: both our SRU language modeling and CIFAR image classification experiment have included weight decay in our default model, and applying activation norm penalty still obtains improvements.

For our LSTM experiments, we apply weight decay to all weight matrices except for the word embedding and the final linear projection.

| Model | $\|\phi(x)\|_2$ |
|---|---|
| LSTM | 19.8 |
| LSTM + WD | 19.4 |
| LSTM + ANP | **9.7** |
| VD-LSTM | 17.9 |
| VD-LSTM + ANP | **8.1** |

(a) The $L_2$ Norm of the final output of LSTMs.

| Model | Validation |
|---|---|
| $\eta_w = 10^{-5}$ | 84.3 |
| $\eta_w = 10^{-4}$ | 84.2 |
| $\eta_w = 10^{-3}$ | 84.3 |
| $\eta_w = 10^{-2}$ | 83.5 |

(b) Weight decay's effect on perplexity in PTB

Table 6: Exploring the difference between activation norm penalty, weight decay, and variational dropout. (a). LSTM+WD refers to LSTM with weight decay at $\eta_w = 10^{-2}$; (b). Weight decay is applied to medium-sized LSTM with feedforward dropout on Penn Treebank

From previous literature (Achille & Soatto, 2017) and our derivations, we know that weight decay, dropout, and activation norm penalty all reduce correlation between input and output. From Table 6 (a), we can see that all three methods decrease the $L_2$ norm of the output to various extents. We also show that in Table 6 (b), when applied to vanilla LSTM with feedforward dropout, weight decay and activation norm penalty have different effects on final task performance as well.

## 7   CONCLUSION

In this paper we present a simple way to extend information bottleneck principle to the training of recurrent neural network. We demonstrate how information bottleneck principle can be applied not only to Bayesian neural networks, but to general neural networks with dropout. We derived the activation norm penalty from the variational dropout framework, and observe consistent improvements to state of the art architectures when applied to language modeling and image classification.

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
