# OpenReview forum: "From Information Bottleneck To Activation Norm Penalty"
_ICLR.cc/2018/Conference — Reject_

### Official Review · AnonReviewer2 · 2017-11-24
**L_2 regularization of activations with theoretical underprinnings**

**Rating:** 7
**Confidence:** 3

**Review:**

The paper puts forward Activation Norm Penalty ("ANR", an L_2 type regularization on the activations), deriving it from the Information Bottleneck principle. As usual with Information Bottleneck style constructions, the loss takes on a variational form.

The experiments demonstrate small but consistent gains with ANR across a number of domains (Language modelling on small datasets, plus image classification) and baseline models.

A couple of things that could be improved:

- The abstract claims to ground the ANR in the variational dropout framework. When it is applied without dropout to image classification, shouldn't that be explained?

- Maybe dropping the determinant term also deserves some justification.

- Very recently, Activation Regularization by Merity (https://arxiv.org/abs/1708.01009) proposed a similar thing without theoretical justification. Maybe discuss it and the differences (if any) in the related work section?

- The Information Bottleneck section doesn't feel like an integral part of the paper.

My two cents: this work has both theoretical justification (a rare thing these days) and reasonable experimental results.

There are a number of typos and oversights:

- Abstract: "variation dropout"
- Section 2:
  - x is never used
  - m in b = m + \sigma\epsilon is never defined (is it the x above?)
- In Section 3.2, equation 11 subscript of x_i is missing
- Section 6, Ungrammatical sentence: "Even though L_2 ..."

---

> ### Author Response · Authors · 2018-01-05
> **Re**
>
> Hi,
>
> Thank you for pointing out the connection to Stephen Merity's paper (we have not read that paper).
>
> Application of ANP without dropout is a concern we have also noticed. We chose to share that set of experiments because we think it is important to share with people that empirically we found that ANP works on all settings of the neural network. The theory framework we have chosen (variational dropout) cannot accommodate/explain this setting, but we hope other theoretical frameworks can offer insight.
>
> The editing suggestions are very insightful, and we plan to fully investigate the effect of the log-determinant term (both computationally and theoretically) and offer a penalty that is closer to the true form of IB :)

---

### Official Review · AnonReviewer1 · 2017-11-25
**Important problem tackled, but paper lacks strictness**

**Rating:** 4
**Confidence:** 3

**Review:**

This paper tries to create a mapping between activation norm penalties and information bottleneck framework using variational dropout framework. While I find the path taken interesting, the paper itself is hard to follow, mostly due to constantly flipping notation (cons section below lists some of the issues) and other linguistic errors. In the current form, this work is somewhere between a theoretical paper and an empirical one, however for a theoretical one it lacks strictness, while for empirical one - novelty.

From theoretical perspective:
The main claim in this paper seems to be (10), however it is not formalised in any form of theorem, and so -- lacks a lock of strictness. Even under the assumption that it is updated, and made more strict - what is a crucial problem is a claim, that after arriving at:
tr[ K(X, X) ] - ln( det[ K(X, X) ] )
dropping the log determinant is anyhow justified, to keep the reasoning/derivation of the whole method sound. Authors claim that quote "As for the determinant of the covariance matrix of Gaussian Process, we cannot easily evaluate or
derive its gradient, so we do not include it in our computation." Which is not a justification for treating the derivation as a proper connection between penalising activities norm and information bottleneck idea. Terms like this will emerge in many other models, where one assumes diagonal covariance Gaussians; in fact the easiest model to justify this penalty is just to say one introduces diagonal Gaussian prior over activations, and that's it. Well justified penalty, easy to connect to many generalisation bound claims. However in the current approach the connection is simply not proven in the paper.

From practical perspective:
Activation norm penalties are well known objects, used for many years (L1 activation penalty for at least 6 years now, see "Deep Sparse Rectiﬁer Neural Networks"; various activation penalties, including L2, changes in L2, etc. in Krueger PhD dissertation). Consequently for a strong empirical paper I would expect much more baselines, including these proposed by Krueger et al.

Pros:
- empirical improvements shown on two different classes of problems.
- interesting path through variational dropout is taken to show some equivalences

Cons:
- there is no proper proof of claimed connection between IB and ANP, as it would require a determinant of K(X, X) to be 1.
- work is not strict enough for a theoretical paper, and does not include enough comparison for empirical one
- paper is full of typing/formatting/math errors/not well explained objects, which make it hard to read, to name a few:
 * fonts of objects used in equations change through the text - there is a \textbf{W} and normal W, \textbf{I} and normal I, similarly with X, Ts etc. without any explanation. I am assuming font are assigned randomly and they represent the same object.
 * dydp in (5) is in the wrong integral
* \sigma switches meaning between non-linearity and variance
* what does it mean to define a normal distribution with 0 variance (like in (1)). Did authors mean  an actual "degenerate Gaussian", which does not have a PDF? But then p(y|t) is used in (5), and under such definition it does not exist, only CDF does.

* \Sigma_1 in 3.2 is undefined, was r(t) supposed to be following N(0, \Sigma_1)  instead of written N(0, I)?

---

> ### Author Response · Authors · 2018-01-05
> **Re**
>
> Hi,
>
> Thank you for your detailed response and a thorough read of our paper. Thank you for recognizing our effort to tackle this problem. As you rightly pointed out, the lack of treatment for the second term (log-determinant) is unsatisfying. We cannot show whether ANP is an upper/lower bound of IB objective nor can we prove that ANP \prop IB.
>
> There are two approaches to integrate IB objective to neural network:
> 1). Weaken the architecture: Alemi, et al. weakened to a diagonal Gaussian distribution. Similarly, we could choose to output a triangular matrix as output of RNN. We chose not to pursue this route because we are aiming to get SOTA performance on all our tasks. Weakening the architecture capacity (or altering SOTA architecture) will not give us comparable performance. In fact, we did get improvement on all SOTA architectures (in a similar setting).
>
> 2). Weaken the objective: ignore the log-determinant term, resulting in a weaker theoretical claim, but it is simpler to train and optimize and has a wider applicability for all architectures. We chose this path instead.
>
> We do plan to tackle the log determinant estimation problem fully, and investigate the effect of it (both in terms of performance improvement and runtime increase). We appreciate your comment, editing suggestions. We do not think having more empirical results is entirely necessary as language modeling and image classification are from two important domains and already heavily optimized by the community.

---

### Official Review · AnonReviewer3 · 2017-11-29
**weak paper, needs major revision**

**Rating:** 4
**Confidence:** 4

**Review:**

This paper proposes an L2 norm regularization of the output of penultimate layer of the network.  This regularizer is derived based on variational approximation to the information bottleneck objective.

I’m giving this paper a low rating for the following main reasons:

1. The activation norm penalty is derived using an approximation of the information bottleneck Lagrangian term.  However, the approximation in terms of a KL divergence itself contains two terms (Equation 10) and the authors ignore one of those (log-determinant) since it is intractable.  The regularizer is based only on the other term.  Dropping the log-determinant term, thus quality of the resulting regularizer, is not justified at all.  It is just stated that we can not easily evaluate the log-determinant term or its gradient hence it is being dropped.

2. The paper is not very well written, contains errors, undefined symbols and loose sentences, which make it very hard to follow.   For example:
i) Eq. 1: 0 \cdot I_N … what is meant by this operation is not stated, also c_n not defined
ii) “The prior of weight vector … with probability p” … not sure what it means to have a Gaussian mixture with probability p.
iii) “q”, “m” not defined in Eq. 2
iv) Eq. 5 seems broken, the last integral has two dydt terms, also the inequalities in that equation seem incorrect.

Authors show a good gain on two language modeling tasks, CIFAR-10, and CIFAR-100.

---

### Decision · Program_Chairs · 2018-01-29
**ICLR 2018 Conference Acceptance Decision**

**Decision:**

Reject

**Comment:**

All reviewers have acknowledged that the proposed regularization is novel and also results in some empirical improvements on the reported language modeling and image classification tasks. However there are serious concerns on writing and rigor (reviewers Anon1 and Anon3) of the paper. The authors have not uploaded any revision of the paper to address these concerns.